# Miniature Deformable MEMS Mirrors for Ultrafast Optical Focusing

**DOI:** 10.3390/mi14010040

**Published:** 2022-12-24

**Authors:** Afshin Kashani Ilkhechi, Matthew Martell, Roger Zemp

**Affiliations:** Department of Electrical and Computer Engineering, University of Alberta, Edmonton, AB T6G 1H9, Canada

**Keywords:** micro-mirrors, MEMS, fabrication, CMUT

## Abstract

Here, we introduce ultrafast tunable MEMS mirrors consisting of a miniature circular mirrored membrane, which can be electrostatically actuated to change the mirror curvature at unprecedented speeds. The central deflection zone is a close approximation to a parabolic mirror. The device is fabricated with a minimal membrane diameter, but at least double the size of a focused optical spot. The theory and simulations are used to predict maximum relative focal shifts as a function of membrane size and deflection, beam waist, and incident focal position. These devices are demonstrated to enable fast tuning of the focal wavefront of laser beams at ≈MHz tuning rates, two to three orders of magnitude faster than current optical focusing technologies. The fabricated devices have a silicon membrane with a 30–100 μm radius and a 350 nm gap spacing between the top and bottom electrodes. These devices can change the focal position of a tightly focused beam by ≈1 mm at rates up to 4.9 MHz and with response times smaller than 5 μs.

## 1. Introduction

Optical focus control is used in a vast variety of optical systems from imaging devices (such as camera modules) to medical diagnostic and therapeutic instruments. Fast microscopy methods such as optical coherence tomography and nonlinear microscopy require optical focusing, but the performance is limited by the slow response of the focusing components. In most designs, focusing is achieved with lens–motor combinations, but the need for compact and faster components requires new techniques.

Deformable focus control components are an acceptable alternative for the traditional lens–motor techniques [1,2,3]. Examples include liquid-filled lenses [4], electro-wetting lenses [5,6], and deformable mirrors [6,7,8,9,10,11,12,13,14,15,16,17,18,19,20,21]. MEMS-based variable-focal-length lenses use electrostatic forces or pressures to change the focal length of membranes or polymers. Liquid deformable lens systems additionally use electro-wetting or other principles for actuation. These devices, compared to mechanical lenses, have shorter response times and require smaller spaces to operate. However, sensitivity to environmental shocks and ambient temperature limit their potential for many applications.

Aside from such deformable lenses, MEMS mirrors with central deformation have several compelling properties. Sub-millisecond response time, parabolic membrane deformation capable of providing aberration-free focusing, and minimization of moving parts qualify them for numerous applications [22]. For instance, camera modules utilizing deformable mirrors for controlling the focus are small, fast, and less power-hungry than competing voice-coil focusing systems.

Himmer et al. [2] designed and fabricated a silicon nitride deformable mirror with a spherical aberration adjustment for controlling focus. The mirror had a 1 mm diameter with a maximum ≈800 nm stable central deflection. The deflection effectively changes the focal length of the mirror from 36 to 360 mm. The mirror is fabricated by depositing a low-pressure chemical vapour deposition (LPCVD) nitride thin-film as a membrane on a thin phosphorous silicate glass (PSG) sacrificial layer, then selectively etching the PSG and extra etching of the substrate with tetramethylammonium hydroxide (TMAH) to form a gap. A simple fabrication process and high optical power are some advantages of this design. However, LPCVD nitride has a high surface roughness, which can increase the scattering of the reflected light. Other groups have fabricated similar devices, but they still exhibited non-ideal surface roughness.

Polymer-based materials have been considered as an alternative to nitride membranes. These materials have several advantages, such as the possibility to form large membranes, a smoother surface, a simpler fabricating process, and larger membrane strokes [2]. Lukes et al. [1] developed two different fabrication techniques for fabricating deformable mirrors with a polymer membrane. Using SU-8 as the membrane material and silicon as the sacrificial layer, they achieved an acceptable membrane surface roughness. The mirrors had a central deflection of 16.7 μm (in the stable range) and a frequency response up to 1 kHz. Sacrificial etch holes occupied 3% of the membrane and increased the scattering of the reflected rays.

Hsieh et al. [7] presented a deformable mirror for an auto-focusing camera module by using a polymer-based bonding technique. This technique eliminated the need for placing etch holes on the membrane and improved the surface quality of the mirror. The fabricated mirror had a 32 μm gap, capable of providing stable membrane deformation up to 20 dioptres.

Kamel et al. [23] recently published a resonant adaptive deformable mirror based on MEMS technology. Their designed deformable mirror has a 1.6 mm circular plate suspended over 49 electrodes with an air gap and actuates electrostatically. Electrodes are arranged into four concentric circles, which allows them to excite the membrane into axisymmetric and circumferential modes. Based on their experimental data, their deformable mirror presented resonant frequencies up to 330 kHz, depending on the mode.

Most fast-focusing systems have been limited to kHz focusing rates. However, many emerging technologies could benefit from even-faster focusing. These applications may include laser modulation systems, optical switches, laser printing and marking systems, projection systems, and microscopy systems.

Some microscopy methods use tunable optics for improved depth scanning [22]. Microscopy techniques such as photoacoustic imaging [24,25,26,27], confocal microscopy, and optical coherence tomography (OCT) [28,29] may benefit from faster deformable mirrors. These systems scan light in tissue and record the backscattered light or acoustic waves to generate images. Deformable mirrors can control the focus of the light and improve the depth of focus. Yang et al. [22] introduced a silicon nitrite deformable mirror with an 8 kHz axial scan rate for a Doppler optical coherence tomography (DOCT) system to dynamically adjust the optical beam focus. The DOCT images were taken at one frame/second. Significantly greater frame rates are currently possible with OCT, but not yet using fast focusing systems to improve the depth of focus. Similar resolution and image capturing improvements may be possible with other forms of optical microscopy when using fast-focusing optics, but current dynamic-focusing optics are too slow for many such applications.

Most previous deformable optics and varifocal MEMS mirror systems used collimated beams and relied on membrane deformation to optically focus incidence parallel rays. The use of collimated beams necessitated large membranes, resulting in slow speeds. In contrast, our approach instead uses beams focused on or close to ultra-miniature membranes, similar to capacitive micromachined ultrasound transducers [30,31,32,33,34,35,36], to change the focal wavefront curvature of tightly focused beams to steer the downstream optical focus. This is performed in a way that cannot be described using optical ray-tracing in the sense that focused rays converging to a point will experience a locally flat surface regardless of the curvature. We used small reflective membranes to introduce near-parabolic focal curvature modulation. Because this can be performed with very small membranes, very fast focusing speeds are possible. Our devices can achieve a tight radius of curvature as small as 3mm, leading to fast focal wavefront shaping. We introduce the theoretical analysis comparing scenarios where the incident beam is either focused right at the membrane surface or where the beam is focused slightly away. We found that the optimal incident focus is at a distance equal to the radius of curvature at maximum deflection and show that focal shifts of nearly 10-times the Rayleigh range are possible.

We call our micro-scale deformable mirrors capacitive micromachined optical focusing MEMSs (CMOF-MEMSs). The presented CMOF-MEMSs have a radius of 30 μm and are capable of operating up to 4.9 MHz. The mirror has a surface roughness less than 0.39 nm (close to the ideal of atomic smoothness), minimizing light scattering to a negligible level. The presented mirror has a 100 nm central deflection, which is small (the bigger the deflection, the more the focal shift), yet still effective in creating relative focal shifts of more than 10-times the Rayleigh range. In order to show the effectiveness of our design, we fabricated the mirrors with a silicon-on-insulating-wafer bonding technique and tested the devices in an optical setup to observe the focal length shifts provided by the mirrors. With an optical relay system, we could successfully change the focus of a laser beam up to 8 mm with only a 100 nm deflection of the membranes. With tighter optical focusing, we can achieve 1.3 μm spot sizes and achieve a 300 μm focal length shift.

## 2. Materials and Methods

### 2.1. Principle of Operation and Simulations

The presented CMOF mirrors in this paper have a thin membrane, which can be electrostatically actuated to deform the mirrored membrane, changing the optical power of the device. A simple optical setup to change the focus of a laser beam is illustrated in Figure 1.

As shown in Figure 1, a laser beam is focused on a CMOF mirror through a beam splitter. When electrostatic force is applied, the membrane’s inward deflection forms a parabolic profile, affecting the reflected laser beam wavefront radius of curvature. The surface profile of a CMOF mirror is given from clamped deformable plate theory in [37]:(1)$$h\left(r\right)=h_{0}{\left(1-{\left(\frac{r}{a}\right)}^{2}\right)}^{2}$$
where *r* is the radial distance from the centre of the mirror, $h_{0}$ is the central deflection of the membrane for a given voltage, and *a* is the radius of the mirror. The profile approximates a parabolic mirror near the centre. The radius of curvature given a height function h(r) is given as:(2)$$ROC\left(r\right)=\left|\frac{{\left(1+{h^{\prime }}^{2}\right)}^{3/2}}{h^{\prime \prime }}\right|$$
where a prime denotes a derivative with respect to *r*.

Thus, the radius of curvature (*ROC*) of the mirror at its centre is given by:(3)$$ROC=\frac{a^{2}}{4h_{0}}$$

The focal length of this mirror is given as f=ROC/2. The optical power (OP) of the mirror is:(4)$$OP=\frac{1}{f}$$
typically reported in dioptres (1/m).

Figure 2a shows the OP of a CMOF for different membrane sizes and central deflections. This is obtained using Equations (1)–(4). The simulation results were achieved with MATLAB (Mathworks Inc., Portola Valley, CA, USA). Membrane deflections up to 200 nm were simulated here. When fabricating these devices, there are some tradeoffs. The smaller the membrane, the faster the response is (which is the intent of this paper), but then, large deflections are difficult to achieve with modest (<100 V) voltages. In our fabricated devices, we used gap heights of 350 nm with maximum pre-pull-in deflections of ≈150 nm. As illustrated, the OP for a 30 μm membrane with a 100 nm maximum central deflection is ≈900 dioptres. This OP is substantially higher than the conventional deformable mirrors fabricated with MEMS technology, which is up to 20 dioptres for polymer membranes [7].

To describe the focusing of a laser beam using our proposed devices, we considered a Gaussian beam with waist $w_{0}$ focused on a CMOF mirror. The position of the refocused beam $s^{\prime }$ is given by [38]:(5)$$\frac{1}{\left(s+{\left(z_{R}/M^{2}\right)}^{2}/\left(s-f\right)\right)}+\frac{1}{s^{\prime }}=\frac{1}{f}$$
where $z_{R}=\pi w_{0}^{2}/\lambda$ is the Rayleigh range, $M^{2}$ is the beam quality factor, f=ROC/2 is the focal length of the mirror, and *s* and $s^{\prime }$ are the object and image distances, respectively.

We chose to analyse two cases: (1) when the focus is right on the surface of the undeflected membrane and (2) when the incident focal waist is not on the membrane.

Let us consider Case (1) first: If the focus were right on the membrane, so that the object position s is zero, then the image position, i.e., where the focus occurs after reflection, is given as
(6)$$s^{\prime }=\frac{{\left(z_{R}/M^{2}\right)}^{2}}{{\left(z_{R}/M^{2}\right)}^{2}/f+f}$$

This is zero when the membrane is undeflected. We call the focal shift the difference between the focal position with a deflected compared to an undeflected membrane, which is thus simply $\Delta s^{\prime }=s^{\prime }$. Differentiating $s^{\prime }$ with respect to *f* and setting it equal to zero, we found that the maximum focal shift $s^{\prime }$ occurs when $f=\pm z_{R}/M^{2}$ and that the maximum focal shift in this condition is
(7)$$\Delta s^{\prime }=\frac{z_{R}}{2M^{2}}$$
or half the Rayleigh range when the beam quality $M^{2}=1$. This is not in and of itself a significant focal shift; however, the addition of an optical relay lens can improve the relative focal shift as will be shown below.

Figure 2b,c illustrate the focal length shift and spot size change of a laser beam for membrane radii ranging from 20 to 50 μm with a central deflection from zero to 200 nm. These simulations were performed for the beam waist focused onto the membrane with s=0 and $w_{0}=0.4a$. As we can see, the change of the laser beam focal position for a perfect Gaussian beam with $M^{2}=1$ is ≈1 mm. This change can be further magnified with an optical relay system. Figure 2d illustrates the focal length shift after implementing a refocusing lens with $f_{R}=3$ mm, positioned d=3.2 mm away from the CMOF-mirror. This magnifies the focal length shift nearly two-times the minimum Rayleigh range after the relay lens. The calculations were performed without considering the aberration introduced by the mirror. These simulations also have limitations in that they do not account for the optimal placement of the focal waist and do not explore different beam spot sizes on the membrane with the needed beam capture constraints; however, these limitations will be addressed below and will result in greater relative focal shifts.

Next, consider that the optimal focal shifts may be achieved when the beam waist is not focused on the membrane. In the subsequent analysis, for simplicity, we will not consider the additional relay lens. The maximal relative focal shift that can be achieved is non-trivial and subject to a multi-dimensional optimization problem. We approached this computationally.

Numerical simulations were performed based on the Gaussian beam focusing model presented in Self using $M^{2}=1$. For both the deflected and undeflected membrane cases, the position of the reflected beam waist was determined as $s^{\prime }=f+\frac{s-f}{{\left(\frac{s}{f}-1\right)}^{2}+{\left(\frac{z_{R}}{f}\right)}^{2}}$ (Equation (Equation 9) in Self). The magnification imparted by the reflection is given by $m=\frac{1}{\sqrt{{\left(1-\frac{s}{f}\right)}^{2}+{\left(\frac{z_{R}}{f}\right)}^{2}}}$. The new beam waist and Rayleigh range of the reflected beam were determined as $w_{0}^{\prime }=mw_{0}$ and $z_{R}^{\prime }=m^{2}z_{R}$, respectively. From these properties, the complete beam radius profile of the reflected Gaussian beam can be constructed as $w^{\prime }\left(z\right)=w_{0}^{\prime }\sqrt{1+{\left(\frac{z-s^{\prime }}{z_{R}^{\prime }}\right)}^{2}}$. The focal shift resulting from deflecting the CMOF membrane is defined as $\Delta s^{\prime }=s_{undeflected}^{\prime }-s_{deflected}^{\prime }$, where f→∞ for the undeflected case. The corresponding normalized focal shift:(8)$$\frac{\Delta s^{\prime }}{z_{R}}=\frac{f}{z_{R}}\frac{\left[{\left(s/f-1\right)}^{2}+{\left(z_{R}/f\right)}^{2}\right]\left(s/f+1\right)+\left(s/f-1\right)}{{\left(s/f-1\right)}^{2}+{\left(z_{R}/f\right)}^{2}}$$
was investigated. Figure 3 shows the plots of $\Delta s^{\prime }/z_{R}$ as a function of s/f. Unlike $s^{\prime }$, which possesses a global maximum when $s=f+z_{R}$, $\Delta s^{\prime }/z_{R}$ is more complex. This plot does not tell the whole story, however, since not all combinations of incident focal positions and beam parameters may be accommodated by a mirror of a certain small size.

In determining the maximal focal shift, the problem must be constrained such that the incident Gaussian beam is restricted to a fraction of the membrane radius. For an incident beam with a waist located at *s* and radius profile $w\left(z\right)=w_{0}^{\prime }\sqrt{1+{\left(\frac{z-s}{z_{R}^{\prime }}\right)}^{2}}$, the beam radius on the membrane located at z=0 is given by $w\left(0\right)=w_{0}^{\prime }\sqrt{1+{\left(\frac{s}{z_{R}^{\prime }}\right)}^{2}}$. Accordingly, for the best parabolic performance of the CMOF mirror, we determined the maximal focal shift for the simulated parameters only within a feasible region where the condition $\eta =\frac{w(0)}{a}<0.5$ is satisfied.

Figure 4 shows the representative beam waist profiles for the deflected and undeflected membranes. From this, Δs was calculated for each condition in the parameter space to be optimized. By fixing the membrane radius at 30 μm and the deflection at 150 nm, close to our experimental values, Figure 5 shows the resulting simulations. In this case, the maximum relative focal shift $\Delta s^{\prime }/z_{R}$ was calculated as 16.3. This is quite significant given the small membrane size and minimal deflection, much greater than the 0.5 predicted above when the incident beam was focused directly onto the membrane.

Similar plots of permissible relative focal shifts are further presented in Figure 6 and Figure 7 for the cases of 30 and 300 μm membranes, respectively, for various membrane deflections. Larger deflections and larger membranes thus offer greater relative focal shifts.

In summary, our simulations using Gaussian beam models indicated the potential to achieve appreciable relative focal shifts using small <100 μm membranes and only 100–200 nm deflections. These large shifts were achieved for special combinations of beam placement, beam waist parameters, and membrane deflections and are much greater than the half-Rayleigh range shifts predicted when the beam is focused directly onto the membrane. To the best of our knowledge, this new regime of ultra-small and ultrafast membranes has not previously been explored. Having small membranes will mean unprecedented focusing speeds, which may find advantages in numerous applications.

### 2.2. Device Fabrication

Numerous fabrication processes have been published for fabricating deformable mirrors [1,2]. Silicon nitride deformable membranes have a simple fabrication process, but the membrane size and roughness are limited. Polymer-based deformable mirrors normally have sizes in the range of a few millimetres and exhibit slower speeds compared to rigid membranes such as silicon and silicon nitride. However, they provide a smoother surface finish, resulting in less scattering.

Wafer bonding techniques are extensively used in micro-systems technology [39]. In this process, two wafers are bonded to form the membrane. There have been several bonding techniques introduced for fabricating micromachined capacitive ultrasound transducers (CMUTs) [39,40,41]. Fusion bonding provides a vacuum cavity with a single crystal silicon membrane, which has atomic-level roughness. The presented devices in this paper were fabricated with this technique to minimize the surface roughness and form membranes only tens of microns in size. To the best of our knowledge, this is the first report of using SOI wafer bonding to fabricate such MEMS deformable mirror devices.

The mirror consists of 10 and 100 nm of Cr and Au, respectively, for the reflective coating and top electrode, and a 1 μm single crystal silicon membrane with a 350 nm gap on a <100> low-resistivity silicon prime wafer. The process begins by thermally dry oxidizing a prime wafer to grow 340 nm of high-quality silicon oxide (Figure 8a). This layer is used as an electrical isolation between the top and bottom electrodes and also to form the mirror’s cavity. This layer is important to withstand large electrical fields during unintended pull-in events of the membrane. Dry oxidation has better oxide quality compared to other methods such as plasma-enhanced chemical vapour deposition (PECVD) oxide and wet oxidation. Furthermore, thermally grown oxide has a flat surface profile, resulting in high-yield fusion bonding. A cavity is formed in the oxide layer in such a way that a thin oxide layer is left in the bottom of the cavity. Figure 8b illustrates this step. This is achieved by performing the first lithography process with a positive photoresist (HPR 504) followed by an oxide wet etching process with buffered oxide etch (BOE). The timing of the process is critical to control the etch height of the oxide.

The fabrication was followed by a cleaning process for both the processed prime wafer and a new SOI wafer (which will be bonded on the prime wafer). RCA cleaning is a standard cleaning process for preparing wafers for fusion bonding [42]. The device layer of the SOI wafer was bonded on the prime wafers by using pressure and heat followed by a 1100-degrees-Celsius annealing process. In order to remove the handle layer of the bonded SOI wafer and expose the device layer, a silicon wet etching process was performed with a TMAH solution, while the backside of the prime wafer was protected with a thin PECVD-deposited silicon oxide layer. Figure 8c illustrates the bonded wafer before removing the handle layer. After removing the handle layer of the SOI wafer, the buried oxide (BOX) layer was also removed with a wet etching process with a BOE solution (Figure 8d). During this step, the protective PECVD oxide layer on the backside of the prime wafer was also removed. The device layer of the SOI wafers acts as an etch stop. This was followed by the second lithography step to form the silicon membrane on the cavity. The excess silicon was etched with an inductively coupled plasma-reactive ion etching (ICP-RIE) process (Figure 8e). A third lithography step was followed by a BOE process to create access holes for the bottom electrodes (Figure 8f). Chromium/gold layers were sputter deposited on the wafer and patterned with a fourth lithography step in order to form the mirror’s surface along with the metalization for the connection pads. Figure 8g shows a fabricated CMOF mirror with two electrodes.

## 3. Results

### 3.1. Device Characterization

In order to determine the mirrors’ optical quality, cut-off frequency, surface profile, and capacitance change, several characterization measurements were performed. In the following subsections, the results are presented.

#### 3.1.1. Surface Quality

The surface roughness of an optical component is a key characteristic to minimize the scattering of the reflected/refracted light. In many optical devices, the roughness is required to be less than λ/20, where λ is the optical wavelength [8]. However, given that our membranes are deflecting only tens of nm in our application, a smoother surface may be required owing to the potential wavefront scattering effects of our tightly focused beam spot.

Figure 9a shows the fabricated wafer and the dies of the CMOF cells, and Figure 9b illustrates a helium ion microscope image of a single CMOF membrane. Atomic force microscopy (AFM) (Dimension Edge, Bruker Inc., Billerica, MA, USA) was used to provide the surface roughness measurements. Figure 9c illustrates the results of a test on a 1 μm square area of the membrane. The surface roughness $R_{a}$ was less than 0.39 nm. The peak to peak roughness was also limited to less than 3.6 nm. This low level of roughness was achieved by using a single crystal silicon membrane and carefully controlling the deposition of the metal materials for the reflective coating. The membrane surface itself had a roughness of less than 10 angstroms.

To investigate the optimal metal deposition recipe to achieve the minimum possible roughness, we performed a set of experiments. We deposited aluminium, chromium, and chromium-gold materials with the e-beam (custom-built by Kurt J. Lesker Company, Jefferson Hills, PA, USA) and sputtering techniques (CMS-18, Kurt J. Lesker Company) in various depositing conditions and measured the roughness of each material. In general, the gold deposited on the chromium layer had the minimum surface roughness and adhesion to the silicon. The best depositions were found to be using a 10/100 Cr/Au layer deposited with sputtering under a 7 mTorr chamber pressure, with a 300-Watt forward power, having a deposition rate of 230 pm/s, Washington, DC, USA for both the chromium and gold layers. These settings were equipment-specific and may vary on other machines.

#### 3.1.2. Static Characterization

Surface deflection profiles were obtained with an optical profilometer (ZYGO Optical Profilometer) during the application of the static actuation voltages. The largest stable deflection was recorded to be 160 nm for a 30 μm-radius membrane with an actuation voltage of 29.6 volts. Figure 9d illustrates a 3D profilometer image of a 30 μm membrane. The optical profilometer measurements shown are for a representative deflection of 100 nm. The maximum deflection prior to pull-in would theoretically be expected to be 0.46 of the gap height [37]. Thus, for a 350 nm gap, the maximum deflection was expected to be 161 nm. Experimentally, this varied from device to device, attributable to fabrication non-idealities. Here, the smallest stable deflection without a bias voltage was 20 nm due to atmospheric pressure.

In this design, the effective gap was the combination of a 350 nm empty gap spacing and a 50 nm oxide layer on the bottom. The oxide layer in the gap spacing acts as an isolation layer between the top and bottom electrodes and can withstand up to 50 volts before breaking down. The low actuation voltage minimizes the dielectric charging in the gap isolation material, which may impact device performance and lead to permanent electrostatic pull-in. We tested our fabricated devices hundreds of times in pre-pull-in mode and observed minimal dielectric charging. Figure 10a illustrates the central deflection of a 30 μm membrane with different applied voltages.

The corresponding capacitance change was measured to investigate the charging in the oxide layer. By using a Keithley 4200-Semiconductor Characterization System, the capacitance was measured over hundreds of actuation cycles. Figure 10b presents the acquired data for six such representative consecutive actuations up to the pull-in voltage (the maximum voltage applied was 100 mV less than the pull-in voltage). All the measured tests produced very similar capacitance–voltage curves, indicating that dielectric charging is negligible in pre-pull-in. These curves were highly repeatable over >100 runs as long as the devices were not pulled-in. Pull-in events tended to create dielectric charging, which could modify the actuation trajectories and could be a source of unwanted hysteresis. Both graphs in Figure 10 are normalized to the pull-in voltage, which is 29.6 V.

#### 3.1.3. Dynamic Characterization

The size, thickness, and material composition of the membrane were the main parameters influencing the frequency response of the CMOF mirrors. Previously published mirrors have a response time down to milliseconds [1]. The response time is limited because of the membrane size, which is normally in the mm range. The frequency response of a deformable mirror is inversely related to the membrane size. In this design, we fabricated the smallest deformable membrane reported to date and used a vacuum cavity to avoid squeeze-film damping effects. We carried out the resonance frequency analysis of our designed membranes by performing ANSYS modal simulations, expecting to see the fundamental frequency near 4.56 MHz. In order to validate our simulation data, we measured the resonance frequency with a scanning laser Doppler vibrometer (Polytec MS5000) and a pseudo-random excitation. We recorded the resonance frequency as 4.94 MHz. The small difference between the simulation and the experimental data may be due to fabrication tolerances and membrane thickness variations. The vibrometer results are illustrated in Figure 11a,b.

In order to determine the response time of the CMOF mirrors, a step voltage was applied and the membrane deflection was recorded over time with the scanning laser Doppler vibrometer. Figure 11c shows the time response of a 30 μm-radius CMOF mirror to a step function with a 22 V amplitude. The mirror showed a response time less than 20 μs and settling times less than 5 μs, which is orders of magnitude faster than previously published deformable mirrors. The response time may be even faster than this, but the current measurements were limited by non-ideal voltage steps produced by the function generator, exhibiting overshoot at the beginning and end of each step, lasting about one microsecond.

### 3.2. Focus Control Demonstration

We built a simple optical setup to investigate the focal point shift and Gaussian beam quality degradation with a CMOF mirror. Figure 12 illustrates a drawing of the optical setup. In this setup, we used a diode laser (QPHOTONICS QFBGLD-633-30PM) with wavelength λ = 632.8 nm coupled to a single-mode optical fibre. The single-mode fibre ensures a TEM00 Gaussian beam mode with $M^{2}\approx 1$. The output of the fibre is collimated and telescoped to a desired beam waist size with an FC/PC-connectorized zoom fibre collimator (Thorlabs ZC618FC-A). Then, the beam is focused with the focus positioned close to the CMOF deformable mirror with a long working distance objective lens with a 20× magnification power and numerical aperture of 0.4. Figure 12 shows a camera image of the focused beam on a 30 μm-radius CMOF device.

The beam focus size is important, while the centre of the CMOF MEMS mirror is closer to a parabolic deformation than the edges; therefore, the beam is aligned to have $w_{0}=8$ μm. The reflected beam was focused in front of a Shack–Hartmann wavefront (SWF) (Thorlabs WFS30-7AR) sensor with another objective lens (10X, NA = 0.28, MY10X-803). The SWF sensor was used to measure the spot size and ROC changes ($R_{G}$) of the laser beam. The ROC of the beam as measured by the SWF indicates the desired focal length shifts. To see this, consider that the wavefront curvature of a Gaussian beam is a function of distance *z* from the focus point as given by [43]:(9)$$R_{G}\left(z\right)=z\left(1+{\left(\frac{z_{R}}{z}\right)}^{2}\right)$$

If the observation point *z* is far compared to the Rayleigh range $z_{R}$, then it is safe to approximate the measured $R_{G}$ value as an exact distance of the focus point from the SWF sensor.

To measure the $R_{G}$ value, we used a signal generator with a square waveform to synchronize the actuations of the CMOF mirror with the camera’s trigger. For each bias voltage, the SWF sensor was used to record the ROC value, as presented in Figure 13a. The x-axis is normalized to the pull-in voltage, and the membrane deflection for this measurement was between 18 and ≈45 nm. (The membrane has an 18 nm deflection without any actuation voltage due to the pressure differences between the ambient environment and the cavity.) By implementing another lens in the system and aligning it to have a 7× magnification, we could shift the beam focus by ≈8 mm with a 100 nm deflection of the membrane. Figure 13c illustrates the recorded results with a normalized x-axis to the pull-in voltage. The recorded profile intensity of the Gaussian beam on the SWF sensor for membrane deflection up to 100 nm presents a high-quality Gaussian beam. Figure 13b illustrates the beam intensity profile at the output of the discussed optical system.

## 4. Discussion

The new design of deformable mirrors presented in this work enables ultra-fast laser modulations in the ≈MHz range. This is achieved by controlling the wavefront curvature of the laser beam, focusing the incident beam close to the surface of the membrane and using small highly reflective membranes. The fabrication technique introduced for CMOFs reduces the surface roughness of the mirrors to a few angstroms, mitigating the surface scattering important for diffraction-scale wavefront shaping.

One limitation of our approach is that the maximum focal length shift achieved without additional relay optics is currently about 16-times the Rayleigh range. This could still prove important for fast beam spot shaping and depth scanning in microscopy applications [25]. This is achievable with membranes smaller than most previous varifocal MEMS mirrors, which allows for unprecedented speeds.

Experimentally, it was difficult to achieve the exact placement of the focus at the radius of curvature of the membrane as we predicted would achieve the maximal focal shifts; however, our data demonstrate the proof of principle of fast focal changes.

Our theoretical analysis neglected spherical aberrations due to the membrane shape not precisely following a parabolic focusing profile. Future work should further investigate such aberrations and corresponding effects on diffraction-limited focusing with a high numerical aperture. Our current devices were designed in a regime not quite approaching diffraction-limited focusing, and future work should aim to push the limits in this regime for next-generation microscopy, imaging, spectroscopy, micro-machining, and sensing.

## 5. Conclusions

In this paper, we introduced a new generation of deformable mirrors called capacitive micromachined optical focusing (CMOF) MEMS mirrors, which have several advantages over previous deformable optical systems. Our theoretical analysis provided a new way of investigating the maximum anticipated relative focal shifts for a given membrane and incident beam parameters. Membranes as small as 30 microns, much smaller than previously considered, were predicted to provide relative focal shifts of more than 10-times the Rayleigh range. We used a wafer bonding technique to fabricate ultra-flat mirror surfaces having a single crystal silicon membrane. For the reflective coating of the membrane, we used thin-films of chromium and gold of 10 and 100 nm, respectively. Based on the AFM tests, the surface roughness was measured to be 0.39 nm. This level of roughness has negligible scattering effects on the reflected light and is required for our constructed membranes due to their miniaturized size and diffraction-level effects of the focused laser beams. The temporal step response times were observed to be less than 5 μs. The recorded results including an optical relay showed an ≈8 mm focus shift for only a 100 nm deflection of the CMOF’s membrane. With the unprecedented focusing speeds demonstrated here, new applications to high-speed microscopy, holography, display technology, and laser marking can be envisioned. 

## Figures and Tables

**Figure 1 micromachines-14-00040-f001:**
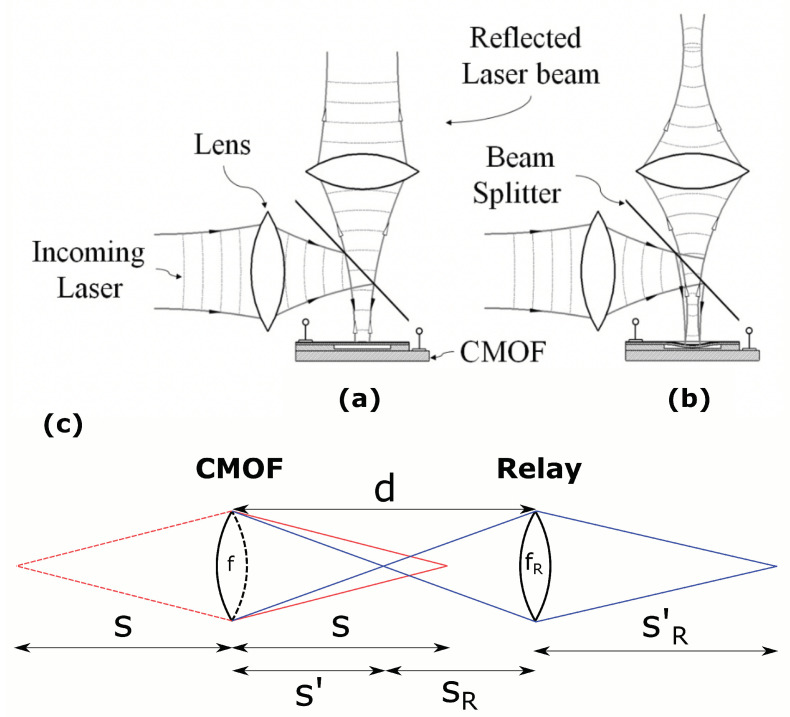
Illustration showing the principle of fast optical focusing with a capacitive micromachined optical focusing (CMOF) deformable MEMS mirror: (**a**) non-deflected CMOF mirror; (**b**) deflected CMOF mirror. (**c**) Illustration of the unfolded geometry of the optical setup. The black dashed line represents the imaginary replacement of a CMOF mirror with a lens. The red solid line represents incident light, and the red dashed line represents the virtual incident light. The blue lines represent refocused light.

**Figure 2 micromachines-14-00040-f002:**
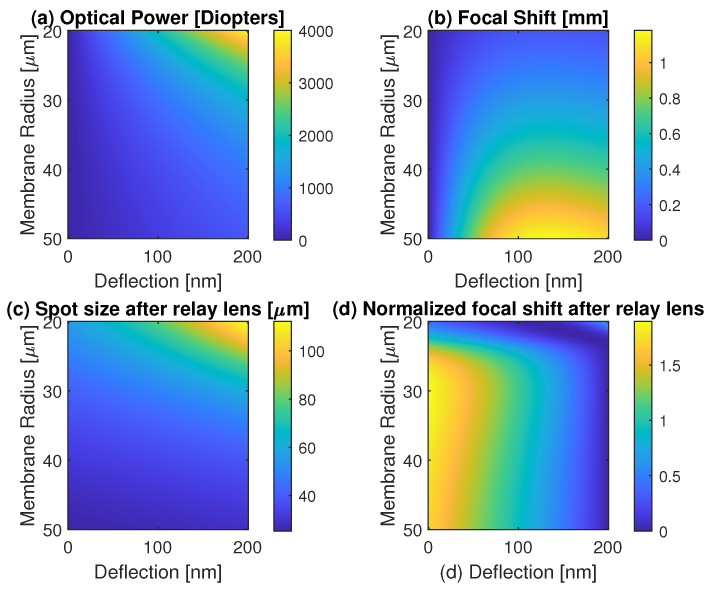
Simulations when the laser is focused onto the CMOF membrane and the Gaussian beam waist is 0.4 of the membrane radius. (**a**) Optical power of a CMOF for various membrane sizes and deflection; (**b**) focal length shift of a laser beam after refocusing with a CMOF for various membrane sizes and deflections; (**c**) focal spot size change of a laser beam after refocusing with a relay lens with a 3 mm focal length; (**d**) focal spot shift of a laser beam after a 3 mm focal length relay lens as normalized by the minimum Rayleigh range of the refocused beam after the relay. In (**c**,**d**), the relay lens is positioned 3.2 mm away from the CMOF membrane.

**Figure 3 micromachines-14-00040-f003:**
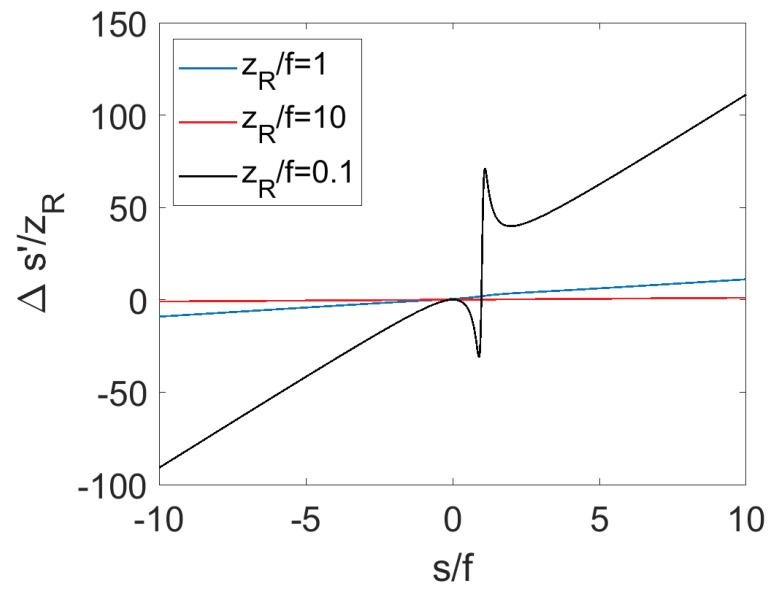
Plot of $\frac{\Delta s^{\prime }}{z_{R}}$ as a function of s/f for three different $z_{R}/f$ values. Larger relative focal shifts are achieved when the focal length is larger than the Rayleigh range of the incident beam.

**Figure 4 micromachines-14-00040-f004:**
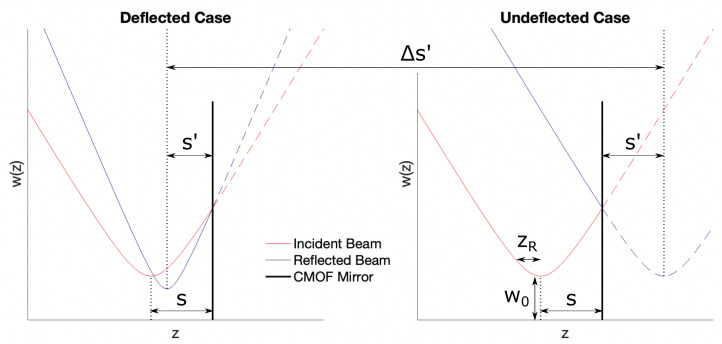
Outline of incident and reflected beam parameters for refocusing of a Gaussian beam by a CMOF membrane. Beam waist simulations for a 30 μm membrane, a 532 nm beam, and a maximum membrane deflection of 150 nm. The plot shows incident and reflected beams, the mirror surface, and the position of the refocusing.

**Figure 5 micromachines-14-00040-f005:**
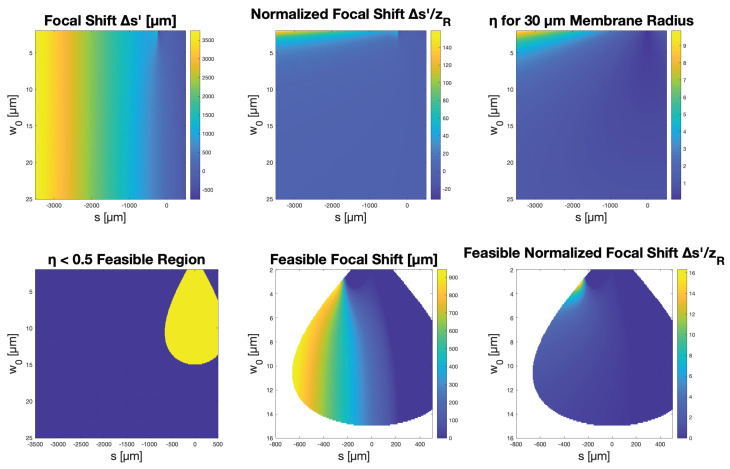
Simulation results for 30 μm membranes and 150 nm deflection using a 532 nm wavelength.

**Figure 6 micromachines-14-00040-f006:**

Relative focal shift simulation results for 30 μm membranes and maximum deflections of (**a**) 50 nm (**b**) 100 nm, and (**c**) 200 nm. The colormaps represent focal shifts normalized by the Rayleigh range of the incident beam.

**Figure 7 micromachines-14-00040-f007:**
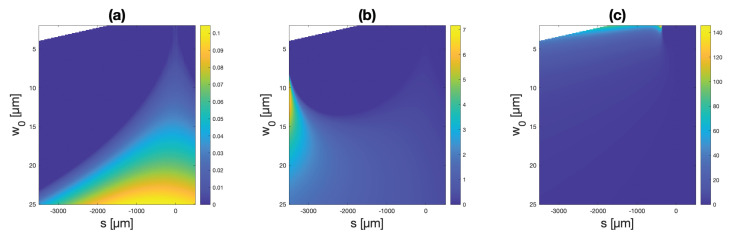
Relative focal shift simulation results for 300 μm membranes and deflections of (**a**) 100 nm (**b**) 1 μm, and (**c**) 10 μm. The colormaps represent focal shifts normalized by the Rayleigh range of the incident beam.

**Figure 8 micromachines-14-00040-f008:**
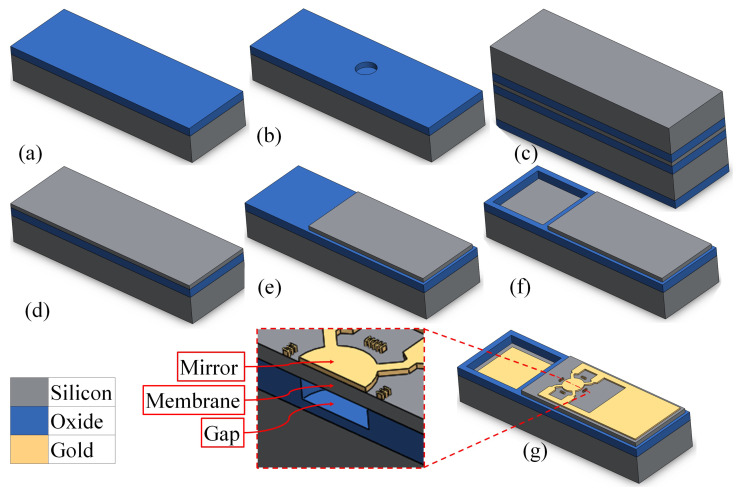
Fabrication process flow: (**a**) dry thermal oxidation to grow 340 nm oxide; (**b**) etching the gap spacing with BOE; (**c**) bonded SOI wafer on the prime wafer, the backside of the prime wafer being protected with PECVD oxide; (**d**) the handle and BOX layers of the silicon wafer are selectively etched, and the device layer is exposed; (**e**) the membrane of the deformable mirror is formed by selectively etching the silicon layer; (**f**) the bottom pad is exposed by selectively etching the oxide layer; (**g**) the metal coating is deposited and etched to form the reflective coating and the electrical pads.

**Figure 9 micromachines-14-00040-f009:**
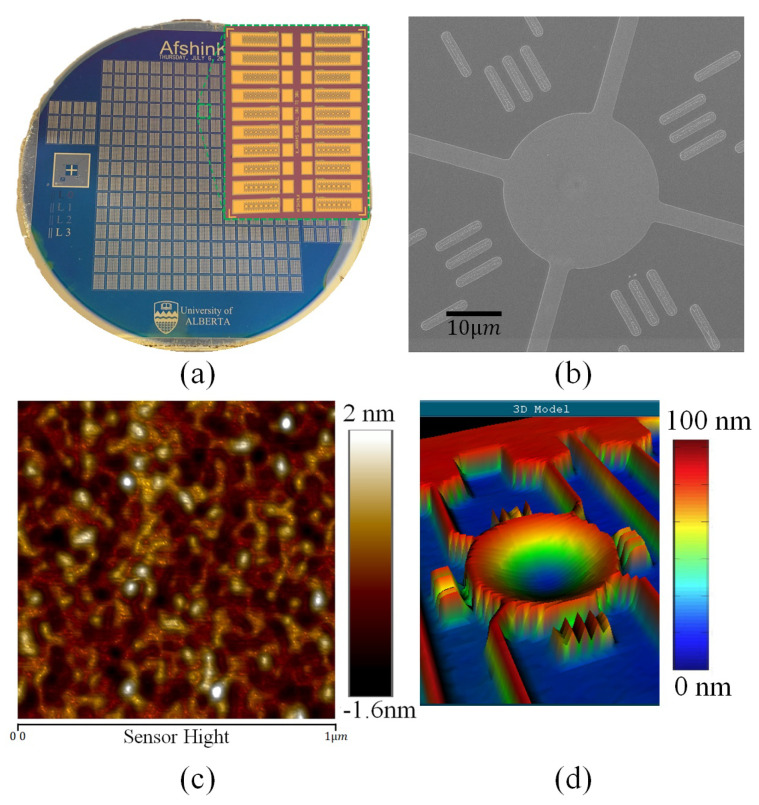
Fabricated CMOF-MEMS deformable mirrors. (**a**) A full wafer view of the fabricated dies; (**b**) helium ion microscopy image of a CMOF-MEMS cell with a 20 μm radius; (**c**) AFM surface profile of a CMOF-MEMS mirror at the centre of the mirror; (**d**) 3D reconstructed image taken by the ZYGO optical profilometer of a CMOF cell with a 30 μm radius and a 100 nm central deflection.

**Figure 10 micromachines-14-00040-f010:**
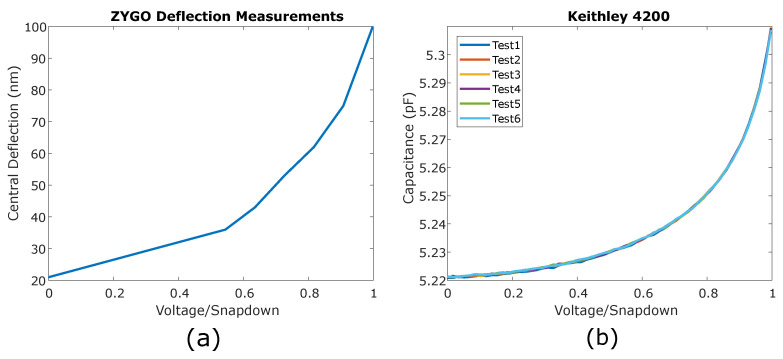
Static tests. (**a**) Deflection versus voltage changes measured by an optical profilometer. (**b**) Capacitive versus voltage changes measured with a semiconductor characterization system. Both graphs are normalized to the pull-in voltage of the CMOF mirror, which is 29.6 V.

**Figure 11 micromachines-14-00040-f011:**
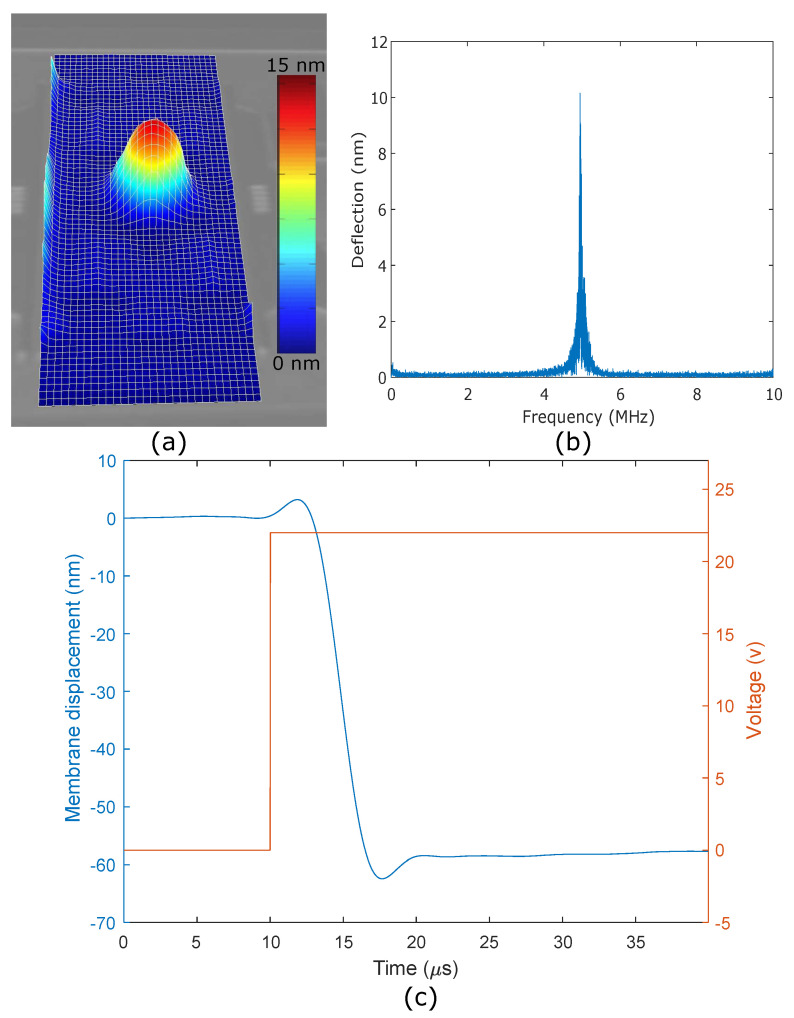
Sample laser Doppler vibrometry measurements. (**a**) The 2D scan showing the membrane displacement for a given frequency. (**b**) Single-point measurement using an 8 V AC signal. The peak shows the first fundamental frequency of the membrane. (**c**) Single-point time domain measurement of the membrane displacement for a step function input with a 22 V amplitude.

**Figure 12 micromachines-14-00040-f012:**
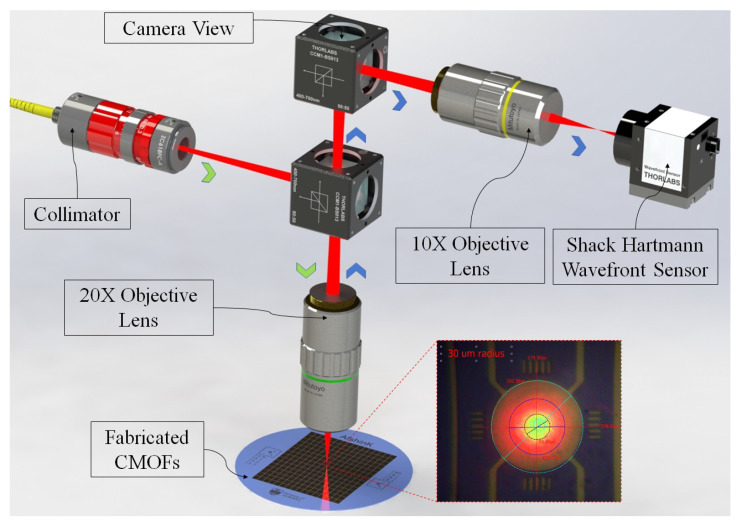
A 3D drawing of the optical setup for testing the CMOFs. The camera-taken picture of the focused laser on a CMOF shows a 16 μm laser spot on a 30 μm-radius CMOF.

**Figure 13 micromachines-14-00040-f013:**
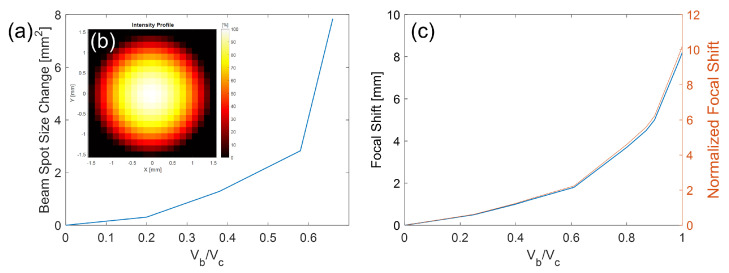
(**a**) Measurements in the beam spot size as a function of bias voltage $V_{b}$ normalized by pull-in voltage $V_{c}$. (**b**) (Inset) Beam intensity profile with the SWF sensor for membrane at 0.7 of the pull-in voltage, associated with a 100 nm deflection. The pixelation of the figure is due to the lens array of the SWF sensor. (**c**) Measured focal point shifts and normalized focal point shifts relative to the Rayleigh range of the refocused beam as a function of the normalized bias voltage.

## Data Availability

Not applicable.

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
