# Peer review of "Miniature Deformable MEMS Mirrors for Ultrafast Optical Focusing"

_micromachines, 2022, doi:10.3390/mi14010040_

Round 1

Reviewer 2 Report

The paper presents an interesting application of very small deformable mirror membranes for ultrafast optical focusing, by using a small mirror close to the focus of focused beams to alter their wavefronts in a more accentuated way. The authors make a thorough description of the complete concept and development, including the theoretical design and determining the expected operation range, describing the fabrication process and mentioning the optimization of the reflective coating process, and the device characterization and demonstration. The authors describe a way to increase focal range tunability by integrating optical relays.

The work is sound, however, the introduction and references are lacking recent works. This makes it hard to validate the author's claim to have produced the smallest membrane reported to date (line 287). The authors should enrich the introduction with more recent works in other capacitive micromachined transducers, deformable membranes and adaptive optics.

Minor comments:

- Introduction: KHz should be corrected to kHz.

- Lines 334-336: Mention to Fig 13 b and c are incorrect with the plots.

- The use of Collapse Voltage in the caption of Figure 13, and Snapdown Voltage throughout the text refer to the Pull-in voltage, which is a more technically correct term in electrostatic actuators that the authors should consider replacing.

- Figures 6 and 7 colorbars and captions do not have units or mention to the relative focal shifts

- In Figure 5, 6 and 7, the only relevant regions of interest are inside the mask (shown in Figure 5 bottom left). An improved plot would remove the data outside this mask, instead of representing the equivalent value of 0 shown in the colorbar (see for example Fig.2 ofhttps://doi.org/10.3390/mi10060366) 
